# First Predation Record of a Tarantula (*Euathlus* sp., Theraphosidae) on a Juvenile Austral Thrush (*Turdus falcklandii*, Turdidae) in Central Chile

**Rubén Montenegro [1,*] and Darko D. Cotoras [2,3]**

[1]  Área de Entomología, Museo Nacional de Historia Natural, Parque Quinta Normal s/nº, Santiago, Región Metropolitana, Chile
[2]  Department of Terrestrial Zoology, Senckenberg Research Institute and Natural History Museum, Senckenberganlage 25, 60325 Frankfurt am Main, Germany
[3]  Department of Entomology, California Academy of Sciences, 55 Music Concourse Dr., Golden Gate Park, San Francisco, CA 94118, USA
*  Correspondence: ramv25@hotmail.com

**Abstract:** We describe the first predation record of an undescribed adult male tarantula from the genus *Euathlus* in a juvenile austral thrush (*Turdus falcklandii* Quoy and Gaimard, 1824) (Aves: Turdidae) in central Chile. The precise moment of predation was not observed; it could have been an opportunistic event or active hunting. In support of the later alternative, this undescribed species of *Euathlus* has been relatively frequently seen walking on tree trunks. These incidental observations raise the question of how important *Euathlus* is as a predator of juvenile birds.

**Keywords:** *Euathlus*; *Turdus falcklandii*; predation; Chile

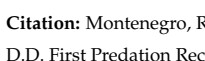



Predation of vertebrates over invertebrates is very common in terrestrial environments. Yet the opposite is rarely observed and reveals poorly understood trophic interactions [1]. Arachnids are among the most common invertebrate predators of vertebrates [1,2]. Some species of mygalomorph spiders capture their prey via ambush from a retreat, while others have a wandering behavior [3]. Although most tarantulas' (Theraphosidae) diets consist of insects, they also include small amphibians, reptiles, birds, and mammals [1,4–9]. Reptiles and amphibians are most commonly mentioned in the literature, with mammals and birds being less frequently reported [10].

During the 19th century, there was a strong debate about whether or not tarantulas prey on birds [11]. Some naturalists accepted it [12–16], while others, mostly influenced by the naturalist and explorer Baron von Langsdorff, rejected it [11,17–19]. The lack of properly recorded examples was the major argument to neglect its existence [20,21]. However, while records of tarantula predation on birds have now been published, this interaction remains relatively uncommon in the scientific literature (reviewed in [10], see also [21,22]).

Here, we describe the predation of a juvenile austral thrush (*Turdus falcklandii* Quoy and Gaimard, 1824) (Aves: Turdidae) by an undescribed adult male tarantula from the genus *Euathlus* in central Chile. The observation was done in an area of the Coastal Ranges, which consists of a mix of native forests, forestry plantations, and agricultural fields. In particular, the event was recorded in Cerro Viejo (33°10′49″ S 71°10′09″ W), located between the communes of Curacaví (Metropolitana region) and Casablanca (Valparaíso region). This observation was done on 16 January 2021 at around 6 p.m. The spider was holding the young bird with the pedipalps and chelicerae. They were located in between the roots of an *Eucalyptus* tree. (Figure 1). The observation lasted for about 10 min. The fact that the bird did not move during that time suggests that it was dead.

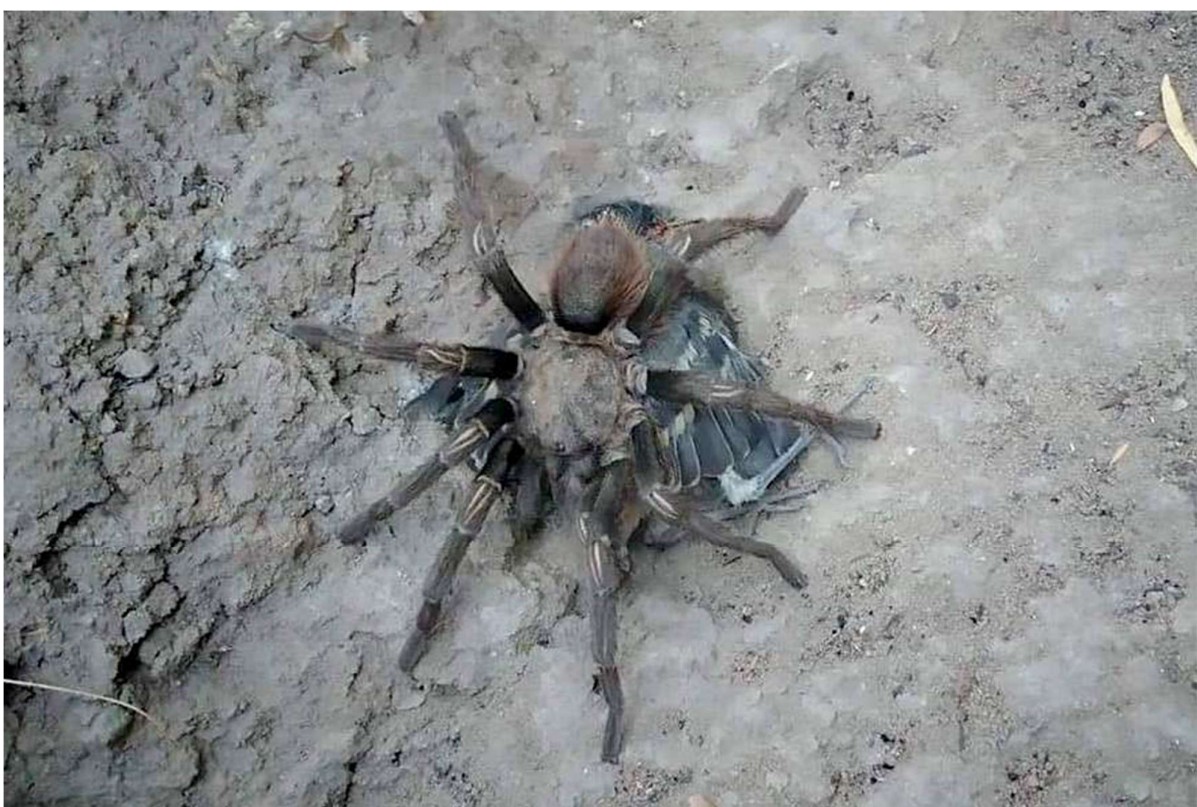

**Figure 1.** *Euathlus* sp. predating on a juvenile austral thrush (*T. falcklandii*). Cerro Viejo, Curacaví, Metropolitana region, Chile. Photo credit: Paula Cárdenas.

No intervention was made and the spider was not collected. However, it was possible to identify the tarantula because one of the authors (R.M.) has studied this population previously. It is possible to distinguish this undescribed species from other *Euathlus* of the country, because of the brownish coloration of the body, two strong red stripes in the patella, and the presence of red setae over the abdomen. This species has an intermediate size (30 mm of total body length) and its distribution includes part of the Valparaíso and Metropolitana regions. It is mostly associated with sclerophyll forest and tend to live under large rocks in small creeks. Sometimes, it could be also associated to the presence of water.

Regarding the biology of the prey, the austral thrush (*T. falcklandii*) is an abundant small to medium size bird with wide distribution along Chile. It is present from Valle de Copiapó to the islands of Cabo de Hornos [23]. This species lives more commonly in semi-open areas, grasslands, forest edges, and open forests, and it also occurs in gardens, parks, and fruit tree plantations [23]. The juveniles are covered in grey feathers with darker tones in the nape and the dorsal area. The edge of the beak has a strong yellow color and the mouth lining is orange on the edges with a yellowish-white color. The legs and the feet have the same color as the body with the nails of an ivory tone [23].

The precise moment of predation was not observed, so for now its nature can only be speculated based on the biology of both species. *Euathlus* is a genus most commonly known by its ground-wandering behavior, therefore the predation on a juvenile austral thrush could have been an opportunistic event. In central Chile, nests with eggs are found from the end of August until the end of January [23], making it possible that the bird was a newly born which fell from the nest and was opportunistically attacked by the spider. However, it is not possible to completely rule out active hunting performed by the *Euathlus* spider. The austral thrushes in central Chile build nests at an average high of 2 m [23]. This undescribed species of *Euathlus* has been seen relatively frequently walking on tree trunks (Figures 2 and 3), in some cases at heights even higher than 2 m (R.M. *personal observation*).

Then, these incidental observations open the question of how important is *Euathlus* as a predator of juvenile birds.

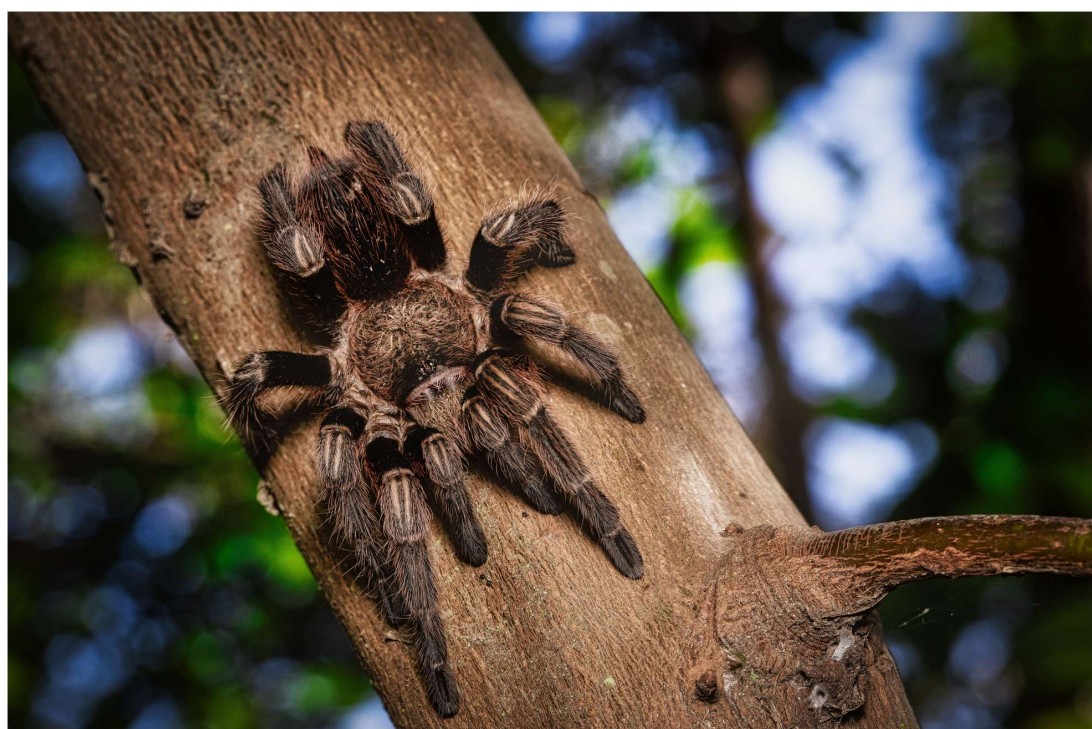

**Figure 2.** *Euathlus* sp. on a tree at 1.8 m high. La Campana National Park, Valparaíso region, Chile. Photo credit: Felipe Rabanal.

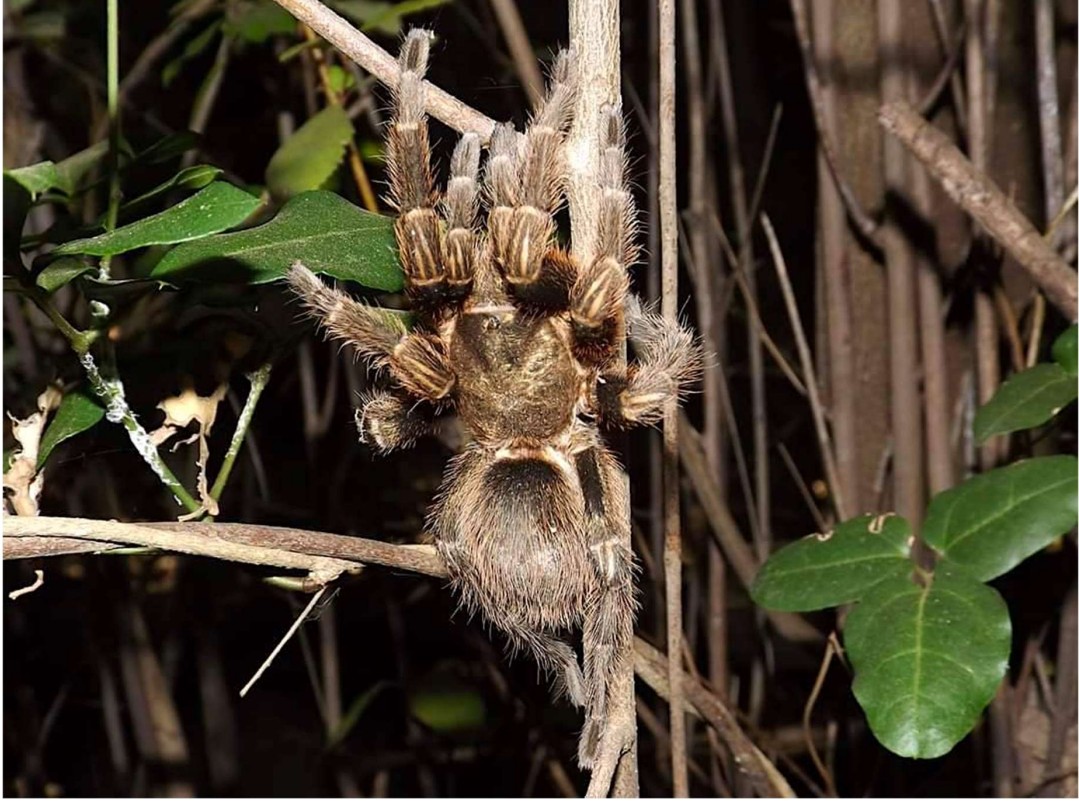

**Figure 3.** *Euathlus* sp. climbing a tree. Lliulliu, Valparaíso region, Chile. Photo credit: Asiel Olivares.

Several records have shown that large theraphosids and even spiders of smaller sizes such as Ctenidae, Lycosidae, and Araneidae, are able to prey on small vertebrates like lizards, snakes, frogs, birds, bats, and mice [1,4,11,24–26]. Frequently these records are associated with tropical areas [1,8,9], being very uncommon reports from temperate locations. Just recently, the first records of vertebrate predation by Chilean theraphosids were presented [2]. They correspond to predation on lizards from the genus *Liolaemus*: (1) the spider *Grammostola rosea* (Walckenaer, 1837) predating on *Liolaemus lemniscatus* (Gravenhorst, 1838), (2) *G. rosea* predating on *Liolaemus tenuis* Duméril & Bibrion, 1837 and (3) *Euathlus* sp. predating on *Liolaemus nitidus* (Wiegmann, 1834). The authors also know about a 2010 sighting of an unidentified tarantula predating on *Liolaemus nigroviridis* (Müller & Hellmich, 1932) (*unpublished data*). So, until now, for Chilean theraphosids, there has been no other record of bird predation.

The documentation of basic aspects of natural history is essential to have a complete understanding of the ecology of the species [21]. In particular, rare observations such as the predation event here presented, allow us to have a full picture of the trophic relationships in a community and estimate how wide-spread these interactions are across the co-occurrent distribution of both participant species (see example in [27]).

**Author Contributions:** Conceptualization: R.M.; writing, review, and editing: R.M. and D.D.C. All authors have read and agreed to the published version of the manuscript.

**Funding:** D.D.C. was supported by a postdoctorate fellowship from the Alexander von Humboldt foundation.

**Institutional Review Board Statement:** Not applicable.

**Informed Consent Statement:** Not applicable.

**Data Availability Statement:** Not applicable.

**Acknowledgments:** We are grateful to Paula Cárdenas, Felipe Rabanal, and Asiel Olivares for providing the pictures for this publication. We also thank three anonymous reviewers.

**Conflicts of Interest:** The authors declare no conflict of interest.

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
