# Peer review of "First Predation Record of a Tarantula (Euathlus sp., Theraphosidae) on a Juvenile Austral Thrush (Turdus falcklandii, Turdidae) in Central Chile"

_diversity, doi:10.3390/d14110946_

Round 1

Reviewer 1 Report

This is a cool record of a tarantula predating on a bird.

The authors says that the bird wasn't moving, there is some evidence to indicate if the bird was alive or dead when they find them?

Author Response

NOTE: All the line numbers refer to the document with Track Changes.

Reviewer 1

This is a cool record of a tarantula predating on a bird.

Montenegro and Cotoras: Thank you for the positive feedback.

The authors says that the bird wasn't moving, there is some evidence to indicate if the bird was alive or dead when they find them?

Montenegro and Cotoras: The evidence that we have that the bird was dead is the fact that it did not move during the 10 minutes of the observation. We have made a more explicit statement now. See lines: 37-38. In addition, the spider was holding the bird with the pedipalps and chelicere, which reinforces the fact that it was already dead.

Reviewer 2 Report

This is a short and interesting description of a tarantula preying on thrush in Chile. Studies like these are important to fully understand the trophic interactions that are not commonly studied or appreciated. Although a nicely written short manuscript, it needs some revisions. The main issue is that the paper has many misspellings that would be easy to identify with a spellchecker. Please use a spellchecker such as Word to avoid these issues.  Additionally, the authors should be more explicit on whether the interaction that occurred was an active predation or scavenger event. For example, was the bird alive and moving, or did it look as if it had previously died and the spider found it? This is a very important distinction. I know it may not be possible, but it is important to give their hypothesis and reasoning.  More specific suggestions:

Title: perhaps change squab to juvenile. I had to look up what this word meant, not sure if it's commonly used.

Line 15. not rare, but rarely observed and not well studied. Need a reference. Cite: 10.1111/geb.13157

Line 19. comma after however. or rewrite to "Although most tarantulas’ (Theraphosidae) diet consists of insects, it can also include...."

Line 19. their is misspelled.

Line 20-21. Cite: 10.1111/geb.13157

Line 21. ", with mammals and birds being less frequently reported."

Line 22. Need a stronger opening sentence. Perhaps something like: "During the 19th century there was a strong debate on whether tarantulas prey on birds" or "Although bird predation was less commonly reported, there was a strong debate..."

Line 25. Should change this sentence. Something like: "However, while records of tarantula predation on vertebrates interaction have now been published, this interaction remains relatively uncommon in scientific literature."

Line 33-34. Should be moved up to the first or sentence of the paragraph.

Line 41. identify is spelled wrong. Also, what is it and what is the population? the bird or the tarantula? Be more specific.

Line 60. locations spelled wrong.

Line 76. important misspelled

Line 82. Not sure how this citation fits with the sentence. Cite the previous example reference as it demonstrates the main points of this sentence.

Author Response

NOTE: All the line numbers refer to the document with Track Changes.

Reviewer 2

This is a short and interesting description of a tarantula preying on thrush in Chile. Studies like these are important to fully understand the trophic interactions that are not commonly studied or appreciated.

Montenegro and Cotoras: We appreciate the fact that the reviewer sees value in our report. We agree that these rare observations are important ecologically.

Although a nicely written short manuscript, it needs some revisions. The main issue is that the paper has many misspellings that would be easy to identify with a spellchecker. Please use a spellchecker such as Word to avoid these issues. 

Montenegro and Cotoras: Thank you for the advice. We have done a new and thorough revision of the writing.

Additionally, the authors should be more explicit on whether the interaction that occurred was an active predation or scavenger event. For example, was the bird alive and moving, or did it look as if it had previously died and the spider found it? This is a very important distinction. I know it may not be possible, but it is important to give their hypothesis and reasoning. 

Montenegro and Cotoras: In the paragraph between lines 208-233 we discuss these two possibilities. On one hand, this could be an opportunistic event due to a bird which felt from the nest and was later consumed. On the other hand, it could be an active predation, which could be supported by the fact that this spider species has a semi-arboreal behavior. See Figures 2 and 3. Note, that a new photograph was added In order to present another and independent record of the tree climbing behavior of the spider.

More specific suggestions:

Title: perhaps change squab to juvenile. I had to look up what this word meant, not sure if it's commonly used.

Montenegro and Cotoras: Thank you for the suggestion. We have implemented the change and further simplify the title.

Line 15. not rare, but rarely observed and not well studied. Need a reference. Cite: 10.1111/geb.13157

Montenegro and Cotoras: We have modified the wording of the sentence as indicated and added the suggested reference. This reference was already in the text, but not supporting this particular statement.

Line 19. comma after however. or rewrite to "Although most tarantulas’ (Theraphosidae) diet consists of insects, it can also include...."

Montenegro and Cotoras: Changed as suggested.

Line 19. their is misspelled.

Montenegro and Cotoras: Sentence removed.

Line 20-21. Cite: 10.1111/geb.13157

Montenegro and Cotoras: Reference added.

Line 21. ", with mammals and birds being less frequently reported."

Montenegro and Cotoras: Sentenced replaced by the reviewer’s suggestion.

Line 22. Need a stronger opening sentence. Perhaps something like: "During the 19th century there was a strong debate on whether tarantulas prey on birds" or "Although bird predation was less commonly reported, there was a strong debate..."

Montenegro and Cotoras: Changed as per suggestion of the reviewer. The first proposed sentence was used.

Line 25. Should change this sentence. Something like: "However, while records of tarantula predation on vertebrates interaction have now been published, this interaction remains relatively uncommon in scientific literature."

Montenegro and Cotoras: Changed as per suggestion of the reviewer. Note that the questionable predation was against birds, not vertebrates in general. So, we used the word “birds” instead of “vertebrates”.

Line 33-34. Should be moved up to the first or sentence of the paragraph.

Montenegro and Cotoras: We moved the sentence about the general area of the observation (ie. Coastal ranges) up in the paragraph. Now, it corresponds to its second sentence.

Line 41. identify is spelled wrong. Also, what is it and what is the population? the bird or the tarantula? Be more specific.

Montenegro and Cotoras: Thank you for detecting the misspelling. It has been changed. We also have clarified that the sentence refers to the population of the tarantula.

Line 60. locations spelled wrong.

Montenegro and Cotoras: Corrected.

Line 76. important misspelled

Montenegro and Cotoras: Corrected.

Line 82. Not sure how this citation fits with the sentence. Cite the previous example reference as it demonstrates the main points of this sentence.

Montenegro and Cotoras: The cited reference corresponds to an example where a rare, but known, spider-frog predatory interaction was recorded in a distant geographic location from where it was first recorded. Therefore, it corresponds to a range expansion of this interaction. It is important to distinguish between species co-occurrence and interaction occurrence. Species interactions many times are context dependent, then it is not necessary a safe assumption that the co-occurrence of a couple of species always implies the existence of a given interaction. This is specially valid in the case of rare interactions.

Reviewer 3 Report

The authors of the manuscript describe a field observation involving a bird-squab being consumed by an undescribed tarantula species. The manuscript is based on a single 10 min long observation and it is documented by a single picture, which is not particularly detailed; the second photograph documents the tarantula species in question climbing a bush.  

The authors extensively review the information about spider predation of vertebrate animals and correctly evaluate its occurrence as rare and/or opportunistic. However, the main issue of the manuscript is that the reported observation does not bring any evidence that the squab was hunted by the tarantula. Spiders are predators, but they would also opportunistically feed on dying/freshly deceased prey, which takes all the novelty from the author’s observation. In fact, the authors do not mention whether the squab was still alive at the time of the observation. The statement: However, it is not possible to completely rule out active hunting performed by the Euathlus spider” is thus completely speculative. The authors claim that one of the authors studied the tarantula “population” in question; therefore, I would expect repeated observations of such phenomenon if the bird squabs were predated upon by tarantula often.   

My second issue with the manuscript is the fact that the authors submit their observation under “Interesting images” category, while they present only 1 picture that illustrates their observation. The picture is not even taken in a full front view; thus, it does not allow to observe the predation in detail. I doubt that anybody, but an arachnologist would be able to tell from the picture that the tarantula is actually feeding on the bird and not just merely sitting on top of it.

Additionally, from a reader’s perspective, the manuscript requires improvements in terms of readability. The paragraphs dedicated to the tarantula ID and the austral thrush do not connect with the remaining text, topics that should go together (e.g. spider ID, its ecology and behavior) are abandoned and re-visited again throughout the text etc.

Although I agree with the authors that field observations are important (including the rare ones) complete our understanding of given subject, overplaying the importance of an observation of undetermined nature does not help the case. I also do not understand what the authors meant by trophic relationships of a species and the “biogeographic distribution of that interaction”. 

Author Response

NOTE: All the line numbers refer to the document with Track Changes.

Reviewer 3

The authors of the manuscript describe a field observation involving a bird-squab being consumed by an undescribed tarantula species. The manuscript is based on a single 10 min long observation and it is documented by a single picture, which is not particularly detailed; the second photograph documents the tarantula species in question climbing a bush.  

Montenegro and Cotoras: The reviewer describes a precise overview of our work. In order to make a stronger case, we have added a second picture independently illustrating the tree climbing behavior of the spider.

The authors extensively review the information about spider predation of vertebrate animals and correctly evaluate its occurrence as rare and/or opportunistic. However, the main issue of the manuscript is that the reported observation does not bring any evidence that the squab was hunted by the tarantula. Spiders are predators, but they would also opportunistically feed on dying/freshly deceased prey, which takes all the novelty from the author’s observation. In fact, the authors do not mention whether the squab was still alive at the time of the observation. The statement: “However, it is not possible to completely rule out active hunting performed by the Euathlus spider” is thus completely speculative. The authors claim that one of the authors studied the tarantula “population” in question; therefore, I would expect repeated observations of such phenomenon if the bird squabs were predated upon by tarantula often.   

Montenegro and Cotoras: The reviewer is correct on saying that we did not have evidence to confirm if the predation event was due to active hunting or opportunistic consumption. Acknowledging that limitation and as the reviewer cited, we discuss both possibilities.

In order to support the idea of active hunting, we provide now two independent photographic records of the tarantula species presenting a tree climbing behavior (originally, we included only one photograph). They complement many additional field observations of the first author.

As for the frequency of this interaction, it can only properly be assessed with systematic surveys. But, we can argue that it is rare as it has never been reported before.

My second issue with the manuscript is the fact that the authors submit their observation under “Interesting images” category, while they present only 1 picture that illustrates their observation. The picture is not even taken in a full front view; thus, it does not allow to observe the predation in detail. I doubt that anybody, but an arachnologist would be able to tell from the picture that the tarantula is actually feeding on the bird and not just merely sitting on top of it. 

Montenegro and Cotoras: We agree with the reviewer that it would have been better to have a series of pictures from different angles. However, only one record was taken and due to its rarity, we think it is worth publishing. Note that the picture is accompanied with our description of the event.

            Moreover, we have two (previously, only one) independent pictures which illustrate the tree climbing behavior of the spider. This is relevant to support the suggestion of a potential active hunting event.

Additionally, from a reader’s perspective, the manuscript requires improvements in terms of readability. The paragraphs dedicated to the tarantula ID and the austral thrush do not connect with the remaining text, topics that should go together (e.g. spider ID, its ecology and behavior) are abandoned and re-visited again throughout the text etc.

Montenegro and Cotoras: In order to improve the readability of the text we have made two modifications. The first was to add an explicit sentence introducing the description of the prey’s biology (Lines 200). The second change was to move the paragraph where we discuss the nature of the predatory event (active vs opportunistic) right after the individual descriptions of the biology of both species. Then, it serves as a synthesis to the information from the previous paragraphs and connects with the following paragraph where other predation events on Chilean theraphosids are discussed.

Although I agree with the authors that field observations are important (including the rare ones) complete our understanding of given subject, overplaying the importance of an observation of undetermined nature does not help the case. I also do not understand what the authors meant by trophic relationships of a species and the “biogeographic distribution of that interaction”. 

Montenegro and Cotoras: It is not our intention to overplay the implications of our observation. That is why we have kept the report short and limited the discussion to only the most relevant aspect (ie. active hunting vs opportunistic consumption).

            Regarding our last sentence, we wanted to make a point on the fact that species interactions many times are context dependent, then it is not necessary a safe assumption that the co-occurrence of a couple of species always implies the existence of a given interaction. In order to clarify that idea, we have re-written the sentence in a more explicit way. Now, it reads: “…and estimate how wide spread are these interactions across the co-occurrent distribution of both participant species [see example in 27].”

Round 2

Reviewer 3 Report

The authors addressed all my comments and concerns and convinced me of the relevance of their findings.